# Scale invariance in kilometer-scale sea ice deformation

Matias Uusinoka<sup>1</sup>, Jari Haapala<sup>2</sup>, Jan Åström<sup>3</sup>, Mikko Lensu<sup>2</sup>, and Arttu Polojärvi<sup>1</sup>

<sup>1</sup>Aalto University, School of Engineering, Department of Energy and Mechanical Engineering, P.O. Box 14100, FI-00076 Aalto, Finland

<sup>2</sup>Finnish Meteorological Institute, Marine Research Unit, Helsinki, Finland

<sup>3</sup>CSC – It center for science Ltd. P.O. Box 405 FI-02101 Espoo, Finland

Correspondence: Matias Uusinoka (matias.uusinoka@aalto.fi)

Abstract. Large-scale modeling of sea ice dynamics assumes scale-invariance that is used to calibrate and validate current models. Validity of this assumption, particularly its lower spatial limit, remains poorly understood. Identifying when, where, and why scale-invariance does not apply is essential for linking meter-scale sea ice mechanics with large-scale sea ice dynamics and climate models. Here we address this challenge by employing unique high-resolution ship radar imagery from MOSAiC expedition in an analysis based on novel deep learning-based optical flow technique. Together these allow capturing sea ice kinematics consistently at unprecedented 20-meter spatial and 10-minute temporal resolutions over an entire winter season and into summer over a 10-kilometer spatial domain. We show that the sea ice within this domain remains largely quiescent for extended periods. During distinct events, a 10<sup>2</sup>-meter lower limit for scale-invariance is observed that endures as the ice cover undergoes seasonal evolution. This threshold remains stable throughout the winter, even as deformation features become more localized and distinct, which suggests an intrinsic mechanical constraint that is invariant under varying external conditions. Once the ice transitions to a floe-dominated configuration in summer, no comparable scaling signature emerges. Our results provide a possible limit under which continuum models fail to capture critical fine-scale processes, highlighting the need for approaches accounting for detailed description of discontinuous spatial and temporal behavior of sea ice.

## 1 Introduction

Since the beginning of geophysical-scale numerical modeling of sea ice, rheological models have been a topic of active discussion. Rheological models describe how sea ice moves, fragments, and deforms under external forces caused mainly by winds, currents, and waves (Leppäranta, 2011). Given the direct connection between the future of sea ice and climate, accurate rheological models are essential for reliable climate change projections. The currently available computational power allows continuum models to simulate sea ice dynamics at very high resolutions. However, it remains an open question whether the continuum models calibrated on fractal properties are valid for relatively small sea ice domains (Hutchings et al., 2024), and what is the lower limit of scale invariance in sea ice dynamics (Weiss, 2017). As the sea ice cover is becoming thinner, more fragmented, and transitions to younger and more mobile ice (e.g. Rothrock et al., 2008; Rampal et al., 2009; Kwok et al., 2013), understanding this lower limit is becoming increasingly critical. A more accurate rheological description able to account for

small-scale processes will be needed to predict future ice conditions with mechanical weakening and increased fracturing. Our work observes this limit and discusses how this limit connects to physical phenomena observed in sea ice.

Our work is based on novel observations of a 10 km × 10 km area of sea ice over a nine-month period. We use unique ship radar observations of ice deformation collected during the Multidisciplinary drifting Observatory for the Study of Arctic Climate (MOSAiC) expedition (Nicolaus et al., 2022). Sea ice deformation is rarely studied at these scales. Conventionally, studies focus on large-scale sea ice deformation observed by using buoy or satellite data (Hibler et al., 1973, 1974; Fily and Rothrock, 1986; Rampal et al., 2008; Hutchings et al., 2011; Marsan et al., 2004; Rampal et al., 2019; Bouchat et al., 2022) or on meters-scale ice failure processes related to ice engineering applications or detailed sea ice mechanics (Sanderson, 1988; Hopkins et al., 1999). Only recent studies have begun to fully explore sub-kilometer intermediate-scale sea ice deformations (Oikkonen et al., 2017; Hutchings et al., 2024), even if detailed insight on ice behavior on this scale is essential for linking meter-scale sea ice mechanics with large-scale sea ice dynamics (Feltham, 2008; Weiss, 2017; Weiss and Dansereau, 2017).

35

We describe the general dynamics of the studied sea ice area over one season, focus on the most active periods of ice deformation, and discuss the scale invariance in sea ice deformation. The latter is the key for discussion on the limit for continuum models for ice and often considered a key for understanding sea ice deformation across scales (Schulson, 2004). This information is essential for understanding sea ice dynamics and for model development (Leppäranta, 2011; Weiss, 2013, 2017). Previous studies have indicated that, although larger deformation occur during storms, sea ice is rarely completely inactive at intermediate-scales (Oikkonen et al., 2017). They also show that ice deformation shows spatial and temporal localization over a wide range of scales following power-law form scaling (Marsan et al., 2004; Rampal et al., 2008; Stern and Lindsay, 2009; Marsan and Weiss, 2010). It is assumed that a rheological model, linking stresses to strains, can be developed for the range of scales where ice deformation remains scale invariant. We employ a novel deep-learning-based optical flow tool (Uusinoka et al., 2025a), which allows full-field analysis of sea ice deformation at 20-meter resolution. The high-resolution data enables us to capture fine-scale deformation features and intermittent behaviors previously unavailable, since conventional methods lead to coarser and less accurate data. High-resolution data is crucial given the emergence of the numerical models that allow simulating hundreds-of-kilometers-scale sea ice domains with meter-scale resolution (Åström et al., 2024; Åström and Polojärvi, 2024).

We present two key findings that are all important to account for in modeling of sea ice. The  $10 \text{ km} \times 10 \text{ km}$  sea ice area we studied experienced three major and several smaller deformation periods occurring during the winter. Based on the major deformation periods, we observe that (1) a possible lower limit for the scale invariance for the studied area was of magnitude  $10^2 \text{ m}$ ; and (2) this limit remained constant for the deformation events until the spring, after which it could not be defined due to the breakup of the ice pack. The observation of highly varying deformation activity differs from the observations of Oikkonen et al. (2017) as their results indicate deformation activity at the intermediate scale to be constantly active although at varying magnitudes. However, their area of measurement was closer to Marginal Ice Zone than that of MOSAiC central observatory. Instead, we find that the deformation events are strongly intermittent and observe multiple periods during the winter season characterized by little activity. (1) is noteworthy, as the limit for scale invariance in sea ice deformation has been discussed for over two decades (Hibler and Schulson, 2000; Schulson, 2004), with authors hypothesizing it might apply at scales comparable

to ice thickness (Weiss, 2013, 2017). The observed deviation from the spatial scaling relation complements the recent work of Hutchings et al. (2024) that pointed out the spatial scaling behavior to vary with spatial scales. We also find that the  $10^2$  m limit is independent of the temporal scale, while the magnitude of error in scale invariance increases with spatial and temporal scale. (2) reflects the change in scale invariance due to seasonal variation in the ice field dynamics, when it transitions from pack ice to floe ice. We are able to capture the lower limit and the seasonal transition, whereas earlier work extending to an average spatial scale of 250 m did not (Oikkonen et al., 2017).

The paper is structured as follows. Section 2 introduces the ship radar data and the tools used in the analysis. Section 3 describes the nine-month-long deformation rate record and highlights the intermittency of the ice deformation events. Then it describes the major deformation events in detail and performs mean-based deformation analysis. Section 4 discusses the implications of our results and their interpretation before Section 5 concludes the paper.

#### 2 Methods

## 2.1 Data

65

Figure 1 shows examples of the radar images used as the basis of our analysis. These were collected by using a radar signal digitizing system similar to Oikkonen et al. (2017) deployed on RV Polarstern as part of the sea ice measurement activities during MOSAiC (Nicolaus et al., 2022). The system continuously collected images of pack ice from mid-October 2019 until the end of September 2020. The original data set included radar images captured at 2.4 s intervals covering a  $20~\rm km \times 20~km$  area with an  $8.33~\rm m$  resolution and oriented along the bow of the ship. The quality of the backscattered radar signal considerably decreased beyond 5 km from the ship, for which we cropped the images to  $10~\rm km \times 10~km$  to ensure consistent estimation quality over the entire image.

The digitized images contained interference from other ship and helicopter radars, which was removed by stacking three consecutive images and retaining the minimum pixel value for each point. Faulty images were identified and removed based

Figure 1. Radar images of the  $10 \text{ km} \times 10 \text{ km}$  sea ice area. Light areas are deformed ice and dark areas level ice and leads. The white squares indicate the different spatial scales, L, analyzed below.

on missing azimuth pulses, spurious rotation-symmetric patterns, incomplete rasterization, or narrow shadowed sectors. The filtered data set has a 1-minute temporal resolution. Despite filtering, the data still contained high-frequency noise concentrated in specific areas of the images. This noise was at least partly due to actual physical phenomena, such as minor vertical displacements caused by ocean swell, small-scale ice deformation, or radar vibrations due to strong winds. The noise manifested as abrupt, high-intensity signal variations, making it particularly challenging to extract consistent displacement estimates from the images. We performed spatial averaging of the images through bilinear interpolation to reduce artificial noise in the radar data, which yielded a final spatial resolution of 10 m.

## 2.2 Recurrent neural network-based optical flow

The ice motion was estimated from the ship radar imagery by using a novel deep-learning-based optical flow tool described in detail by Uusinoka et al. (2025a). The tool is built on Recurrent All-pairs Field Transforms (RAFT) optical flow architecture (Teed and Deng, 2020), enhanced with a temporal multi-resolution tree for increased accuracy in small pixel displacements. We found this tool to overcome the typical challenges involved in the analysis of the radar data while allowing high-resolution full-field analysis of ice motion by providing  $\sim$ 750 000 trajectories for each radar image. The neural network model was fine-tuned using the unique noise maps extracted from the MOSAiC data to further avoid the radar noise affecting the displacement estimates. To further ensure the robustness against the high-frequency noise in the data, described in Uusinoka et al. (2025a), we chose 10-minute intervals between sequential images for sufficient displacements. Based on the sequential data, we integrate the individual displacement fields to 24-hour trajectories. We limit the trajectory length to 24-hours to avoid data loss due to artificial rotations in the radar images.

## 2.3 Strains and spatiotemporal scaling

100

We calculate deformation estimates using quadrilateral cells in accordance to Bouchat and Tremblay (2020) to avoid overestimation of deformation rates due to increased boundary-definition errors (Lindsay and Stern, 2003; Bouillon and Rampal, 2015). Instead of assuming infinitesimal strains, commonly used in sea ice dynamics, we apply Green-Lagrange strains (Appendix A). This choice was made to account for large strains and rigid body rotations; infinitesimal strains are prone to yield erroneous results when these occur (Gurtin, 1982). Strain estimates for large-scale ice drift are around 9 % over 24-hour-period and often higher at smaller scales due to localized ice deformation (Leppäranta and Hibler III, 1987; Hibler et al., 1973). We derive the estimates from finite strains to rotation-fixed infinitesimal strains, l, to relate our results to prior research (Derived in Appendix A).

We use total deformation rate,  $\dot{E}_t$ , to perform spatiotemporal scaling analysis on the ice cover. Localization and intermittency of sea ice deformation has been shown to follow the mean-based power-law forms (Marsan et al., 2004; Rampal et al., 2008)

$$\langle \dot{E}_t(L,T) \rangle \sim L^{-\beta}$$
 and  $\langle \dot{E}_t(L,T) \rangle \sim T^{-\alpha}$ , (1)

where  $\langle \dot{E} \rangle$  is the mean total deformation rate over the entire field, L and T are the spatial and temporal scales, respectively, and the exponents  $\beta$  and  $\alpha$  represent the corresponding spatial and temporal scaling.  $\beta$  quantifies the degree of spatial localization,

ranging from  $\beta = 0$  (perfectly homogeneous viscous-like deformations) to  $\beta = 2$  (deformation localized into a single point). Similarly,  $\alpha$  quantifies the degree of temporal localization, ranging from 0 (continuous flow) to 1 (a single 'event').

Our scaling analysis uses the dispersion method (Rampal et al., 2008; Oikkonen et al., 2017; Rampal et al., 2019), where we extract velocity gradients with increasing spatial and temporal intervals corresponding to specific nominal scales. This extraction is based on the initial coordinates of the corresponding trajectories and involves estimating deformation rates over larger scales to highlight broader patterns. The nominal spatial and temporal scales for computing deformation rates were L = [20, 50, 100, 200, 300, 500, 650, 800] m and T = [10 min, 30 min, 60 min, 3 hours, 6 hours, 12 hours], respectively. It is important that averaged spatial scales below 250 m are considered, as Oikkonen et al. (2017) demonstrate the scale-invariance extending down to this scale.

### 3 Results

### 3.1 Ice cover deformation

Figure 2 shows deformation rate averaged over the  $10 \text{ km} \times 10 \text{ km}$  sea ice area around the MOSAiC central observatory. These records were derived using a 10-minute temporal resolution and a 20-meter spatial resolution and they cover data from

Figure 2. Time series of a) divergence rate,  $\dot{E}_d$ , b) shear rate,  $\dot{E}_s$ , and c) total deformation rate,  $\dot{E}_t$ , calculated with a 10-minute temporal and 20-meter spatial resolution as an average over the deformation field. Note that the y-axis limits are adjusted individually for each subplot to highlight the distinct behavior of the different deformation components and periods.

November 2019 to April 2020. The time series is extended with deformation estimates from July 2020 to include summer conditions in the marginal ice zone (MIZ).

Figure 2 shows that for the most of the winter, the considered area of sea ice experienced only vanishingly small deformation rates. The recurring and extended periods of inactivity are noteworthy, especially the three-week period in February. Only

**Figure 3.** Time series of the three deformation rate components during the three active periods highlighted in Figure 2: a) the November period, b) the January–February period, c) the March period. Additional times series of the first two weeks of July is provided in d) for comparison between different systems. The July case has a larger scale on the vertical axis due to the constant intensity of deformation.

minor shearing could be detected during the quiescent periods. Visual inspection of the radar imagery shows that this low-magnitude shearing resulted from a mixture of subtle ice displacements, rather than significant changes in the ice cover. The high-frequency noise in the radar data may also contribute partially to the low-magnitude shear observed. Further, the figure shows several intermittent deformation events that occurred during three periods of major ice deformation. These periods are highlighted in gray in the figure and occurred in November 2019, January-February 2020, and mid-March 2020. The last one of these is noted for exceptionally high mean deformation rates approaching 0.4 h<sup>-1</sup>. During these three periods, the ice deformed rapidly in various modes, including events with all deformation occurring within few-hour periods. Inactive periods are not seen in July data, which describes ice deformation within the MIZ. In this case the time series describes a significantly different system with practically continuous ice deformation. The deformation rates increase by an order of magnitude compared to winter, and the peak values are much higher than those reported for similar conditions by Oikkonen et al. (2017).



Figures 3a-c provide a more detailed view of three active two-week winter time periods of Figure 2. The three time series exhibit a consistent intermittency similar to that described above for the seasonal analysis. Increased intermittency is observed during the cases of January-February and March (Figure 3b-c) in relation to the observations of Oikkonen et al. (2017), whose results indicated more continuous deformation in conditions closer to the MIZ. For November (Figure 3a), the data shows several deformation events and brief inactive periods, likely due to the low load-bearing capacity of the relatively thin ice (Itkin et al., 2023). In contrast, the active period in February (Figure 3b) is characterized by few intermittent deformation events and long inactive phases, possibly owing to the thickened ice cover. Data for March (Figure 3c) depicts a blend of patterns observed during the preceding months and it is characterized by significant deformations, which suggests a shift from the winter towards the spring. In July (Figure 3d), within the MIZ, ice undergoes constant deformations with few extreme peak deformation events, which in this case signify a uniformly intense deformation across the area studied.

Figure 4. Total deformation (Eq. A5) fields for the 24-hour periods highlighted in Figure 3. Each field show mean deformation for 20 m  $\times$  20 m cells during the period considered. Features exiting the radar images have been neglected and no filtering has been used. In d) all trajectories exiting the radar image during the 24-hour period have been removed.

The deformation fields displayed similar seasonal characteristics, as shown by Figure 4, which visualizes the mean total deformation rate over the 24-hour periods highlighted in Figure 3. Examples of individual 10-minute deformation fields can be found in Uusinoka et al. (2025a). The November case (Figure 4a) has the deformation distributed across several larger scale fractures with numerous smaller deformation features emerging from them and indicating localized activity around them. The February case (Figure 4b) has the deformation concentrated along a line-like lead, with smaller perpendicular deformation features appearing, which indicates secondary failure processes. By March (Figure 4c), nearly all activity becomes localized along a single quasi-linear fracture, representing a large lead. This main feature is flanked by smaller regions where ice fragments collide and form a complex interaction network. Deformation field for July (Figure 4d) is markedly different from the rest, as the entire area studied undergoes viscous-like deformation as it is located in the MIZ during summer conditions.

## 3.2 Spatiotemporal scaling and limit of scale invariance






We conducted a spatiotemporal power-law analysis using the mean total deformation rates of the four 2-week periods high-lighted in Figure 2. We focus the analysis on cases studies from active periods to ensure high enough signal-to-noise ratios. Figure 5a depicts how the scaling exponent  $\beta$  behaved in the case of observations made over various spatial and temporal scales, L and T. Figure 5c shows  $\beta$  as function of T for data with  $L \le 150$  m and  $L \ge 150$  m in comparison to the July case. In Figure 6, similar analysis is performed for the temporal scaling exponent,  $\alpha$ , as a function of L to explore similar limitations of scale invariance in the temporal domain. In the analysis, we use the July case as a reference point for when the continuum approach and fractal analysis are appropriate, given its relatively uniform and diffuse deformation field (Figure 4d). The July case helps in illustrating how the fractal properties diminish when a continuum perspective begins to break down.

Figure 5 shows that  $\beta$  varies between 0.4 and 0.6 for the winter cases when  $L \ge 150$  m and T = 10 min. The  $\beta$  values align well with the observations of increasing  $\beta$  towards smaller spatial scales in Hutchings et al. (2024), although they are about  $\sim 25\%$  smaller than those observed by Oikkonen et al. (2017). Further, the  $\beta$  values decrease with increasing T in all winter cases as expected. For data  $L \ge 150$  m, the two-week period in January showed the highest values of  $\beta$ , even if this period showed the lowest peak values for total deformation rates (Figures 3). Additionally, the November and March periods exhibit nearly identical scaling behavior, even if their deformation time series and deformation fields are not alike (Figures 3 and 4). Values of  $\beta$  for July data are low, which is expected based on the deformation field in Figure 4d.

For scales  $L \le 150$  m, a notable shift in system behavior occurs, with the values of  $\beta$  being about constant for all T used in the analysis. At the smaller spatial scales, mean strain rates are similar across different time scales, suggesting a dominance of the largest deformation features in the overall deformation field. Figure 1 illustrates scales  $L \le 150$  m, which are often less than the typical width of deformation features such as leads. At larger scales of  $L \ge 150$  m,  $\beta$  stabilizes around 0.2 for all winter cases around  $T \approx 1...3$  h, which is the value of  $\beta$  for scales for the scales  $L \le 150$  m for all T. Data for July does not show a scale-depended shift as  $\beta \approx 0.1$  to 0.2 for all T for both  $L \le 150$  m and  $L \ge 150$  m. For  $L \le 150$  m, the values of  $\beta$  in the winter and the July case exhibit comparable magnitudes and similar behavior, suggesting that at sufficiently small spatial scales (or, as seen in Figure 5a, at longer averaging times) with the given spatial domain size of  $10 \text{ km} \times 10 \text{ km}$ , the deformation processes tend to loose the typically observed fractal properties.

From the above, it is clear that the mean-based scaling does not apply for all L, as data for  $L \le 150$  m behave differently than those for the larger scales. Therefore we looked for a critical length scale,  $L_c$ , for the scale invariance, using the data for the case in Figure 5a with a 10-minute temporal resolution. Figure 5b illustrates how this was done. We performed a least-squares linear fit to the data points  $\{(L_j, \dot{E}_j)\}_{j=1}^n$  on a log-log scale. If the coefficient of determination was  $R^2 \le 0.95$ , we excluded the data points with the smallest L, then repeated the fitting with the remaining n-i data points, where i is the number of

Figure 5. a) Mean based spatial scaling properties of total deformation rate (Eq. A5) during the four periods highlighted in Figure 2. The linear fits are for the data with length scale  $L \ge 150$  m. b) Limit for the scale invariance for the four two-week periods of Figure 3 with T=10 min. Coefficient of determination  $R^2$  and scaling coefficient  $\beta$  are for the fit applied on the data points that follow the mean-based scaling (marked with white dots). The dashed line represents the scale of  $L_c$  determined as the point where the relative difference between the fit and the data exceeds 10 %. c) The development of the spatial scaling exponents,  $\beta$ , over different temporal scales. The development of  $\beta$  is considered individually for both  $L \ge 150$  m and  $L \le 150$  m in comparison with the July case.

excluded points. This was continued until  $R^2 \ge 0.95$ . Then  $L_c$  was defined as L where the relative difference between the fit and the data exceeded 10 %.



Figure 5b shows that the magnitude of  $L_c \sim 10^2$  m throughout the winter season, with only minor variations observed between the three winter time periods of active deformation. Based on the deformation patterns of Figure 4,  $L_c$  in November appears to be related to multiple minor deformation features across the whole  $10 \text{ km} \times 10 \text{ km}$  sea ice area, whereas later during the winter, it appears to be related to few major features. We also tested that this estimate is consistent for other deformation events occurring during the wintertime. However, we could not find  $L_c$  for July. This was expected due to the diffuse nature of summer time ice deformation, already seen in the analysis of deformation time series and deformation fields above (Figures 2 and 4).

Figure 6a show the behavior the system over temporal scales, T, and Figure 6b compares the derived values of  $\alpha$  with the summer conditions. Values of  $\alpha$  derived for our data, describing the intermittency of the deformation events, also showed a spatial-scale dependent change. For scales  $L \leq 150$  m,  $\alpha$  was about constant for all winter time cases as Figure 6b shows. This stabilization of  $\alpha$  suggests that to maintain the typically observed scaling behavior, one must consider spatial scales significantly larger than 150 m. For observations with  $L \geq 150$  m, on the other hand,  $\alpha$  decreased with increasing L. July data does not show this feature. Overall, the alpha  $\alpha$  values were about 30% lower than those reported by Oikkonen et al.

Figure 6. a) Mean based temporal scaling properties of total deformation rate similarly to Figure 5. The linear fits over for all the presented temporal scales. b) The development of the temporal scaling exponents,  $\alpha$ , over different spatial scales. The development of  $\alpha$  is presented in comparison with the July case.

(2017), despite our time series exhibiting stronger intermittency. This is likely because of our analysis considering only active deformation periods, where the average deformation rates even over longer temporal scales stay consistently larger.

## 205 4 Discussion





Our study presents new insights into sea ice dynamics by extending the scaling analysis towards smaller scales. We observe a possible limit for scale invariance at approximately  $L_c \sim 10^2$  m for winter pack ice. For  $L_c > 10^2$  m, our results align with previous observations of fractal properties at similar scales (Oikkonen et al., 2016, 2017). Scale invariance at intermediate scales has been thoroughly investigated only by Oikkonen et al. (2017), who analyzed a 15 km  $\times$  15 km with  $L \ge 250$  m and  $T \ge 10$  min, with the results considered evidence for scale invariance extending at least down to L = 250 m (Weiss, 2017; Rampal et al., 2019). Our findings now provide observations on the lower limit for the scale invariance. Since our analysis was conducted on the same  $10 \text{ km} \times 10 \text{ km}$  area over the winter season, it can be argued a different region might have produced a different magnitude for  $L_c$ . With this in mind, it is intriguing that  $L_c \sim 10^2$  m remained constant, even if the ice cover properties went through changes over the winter (Itkin et al., 2023).

We highlight several points of evidence to provide increased certainty to the result of observing a spatial lower bound for scale invariance. The synthetic tests in Uusinoka et al. (2025a) confirm that the neural network optical flow algorithm retrieves accurate pixel displacements even with faced with artificial radar-like noise. Secondly, the signal-to-noise estimates - based on Bouchat and Tremblay (2020) and presented in Uusinoka et al. (2025a, Supp. S2) - demonstrate that localized deformation signals dominate random fluctuations in the data yielding a considerably high ratio even at the finest resolutions. Thirdly, the lower bound consistently appears during active fracturing events in winter pack ice (when signal-to-noise ratios are high), while the summer conditions with similar noise patterns do not exhibit any such cutoff. This would suggest that the observed phenomenon relies on mechanical processes rather than measurement error resulting from radar noise. The observed  $\sim 10^2$  m length scale coincides with a typical width of major leads and shear zones observed in the radar data, giving a potential physical explanation for why scale invariance would break at such limit. Still, it remains possible that the smallest deformation features are indistinguishable in the radar data, which would result in stagnation of the spatial scaling exponent at low scales. Furthermore, the checkerboard pattern of noise seen in Figure 4 around active deformation zones could have an unidentified effect on the power-law scaling. Further error estimation and consideration of the checkerboard pattern is provided in Appendix B. To fully validate the observation of a lower bound for scale invariance, further independent analysis is required. This analysis needs to be based on additional field campaigns, alternative deformation detection methodologies, or complementary datasets.

It is well-established that large-scale sea ice deformation fields are multifractal in both spatial and temporal dimensions (Marsan et al., 2004; Weiss, 2013), which suggests that different statistical moments could reveal further complexities. A multifractal analysis of the MOSAiC ship radar data presented in Uusinoka et al. (2025b) shows a similar limitation for scaling around  $L \sim 10^2$  m over the higher moments where the deformation signal is more distinguishable. The results of Uusinoka et al. (2025b) on multifractal analysis also distinguishes between the early winter deformation fields, characterized by more

continuous spatial and temporal deformation, from the late-winter fields dominated by only a few large deformation features. The results of this paper together with the multifractal observations provide a more complete picture of the scaling relations applicable in ice dynamics.






The variation of the spatial scaling exponent  $\beta$  with increasing spatial scale was recently observed by Hutchings et al. (2024), suggesting that the sea ice deformation rate does not actually exhibit scale invariance. Their analysis considered buoy data down to spatial scales of 1 km, below which they did not have sufficient data points to support claims about smaller scales. By extending these observations to spatial scales smaller than 1 km, we similarly observe a distinct change in the behavior of  $\beta$  at approximately  $L \sim 10^2$  m, thereby complementing the observations of Hutchings et al. (2024). However, in contrast to the buoy data, we observe a clear stagnation in the values of  $\beta$  for  $L \le 10^2$  m. This is likely due to the dominance of larger, distinct deformation features clearly observable with coarser spatial resolutions—mainly leads and shear zones—over smaller deformation features. We also observe that this relationship depends on the temporal scale of the observations. Unlike the buoy data, we have approximately  $4 \times 10^8$  data points over a two-week period at spatial scales of L=20 m and temporal scales of T=10 min. Our results complement the analysis of the buoy data by revealing system regime changes with varying spatial and temporal scales.

Our results have several implications, particularly for modeling sea ice using the continuum approach. Rheological models applied on spatial resolutions larger than  $L_c$  may not capture ice behavior below  $L_c$ , where the continuum assumptions may break down altogether for the ice pack during winter. We see that at the scales studied here, the largest discontinuities dominate the overall deformation field (Figure 4). The size of these features is approximately  $L_c$  (Figure 1). This is particularly pronounced during mid- and late-winter, where the ice field akin to rigid bodies separated by large deformation features. Early winter conditions exhibit simultaneous deformation at multiple locations, resulting in less distinct deformation features. In contrast, the summer ice pack behaved drastically different with no  $L_c$  found, highlighting the need for general rheological models to account for seasonal variation. Moreover, our observations suggest that ice deformation in winter is driven by short-lived (hour-scale) and spatially localized events that generate features on the order of  $10^2$  m. While scale invariance provides a valuable benchmark for large-scale sea ice rheologies, we note that it is not the only measure for model accuracy. Bouchat et al. (2022) show that multiple numerical models can reproduce similar fractal dimensions while diverging in other important ice properties. Our findings should be complemented with other diagnostics and comparisons to ensure a comprehensive model evaluation and development.

How can we model sea ice cover if there is no continuum description at small scales? The discrete element method (DEM) has been used in ice mechanics and dynamics studies on various scales for decades (Hopkins et al., 1991; Hopkins and Hibler, 1991; Hopkins, 1994, 1996, 2004; Hopkins and Thorndike, 2006) and seen as a tool for future studies as well (Blockley, 2020; Hunke et al., 2020). Current DEM developments enable modeling multifracture of sea ice, three-dimensional deformation processes, and interactions of ice features over large sea ice domains, without requiring a continuum model (Manucharyan and Montemuro, 2022; Åström et al., 2024; Åström and Polojärvi, 2024; Muchow and Polojärvi, 2024; Tsarau et al., 2024); apart from that for intact level ice, high resolution DEM tools do not need a rheological model.

### 270 5 Conclusions

Our analysis of high-resolution MOSAiC deformation data observes a 10<sup>2</sup>-m lower bound to the scale-invariance of sea ice deformation. The results challenge conventional assumptions that scale invariance might reach even down to scales comparable to ice thickness and highlights the need for further consideration on the applicable models at smaller scales. We show that sea ice deformation below the 10<sup>2</sup>-m scale is dominated by large features such as leads and fracture zones. These features disrupt the traditional continuum assumption that underlies most high-resolution sea ice models. Seasonal differences further reinforce the complexity of these mechanics: what holds true for winter pack ice breaks down as the ice transitions towards the marginal ice zone in summer. During winter, intermittency and spatial localization in deformation persist, while in summer deformation becomes more granular, defying the applicability of the fractal analysis. The seasonal transition highlights the influence of ice properties, thickness, and external forcing on deformation patterns. Our study was based on unique field observations from ship radar data gathered during MOSAiC expedition and on their analysis by using novel deep learning-based tools; our results demonstrate that such high-resolution empirical data can guide the development of hybrid modeling frameworks that combine continuum modeling with models explicitly accounting for the discontinuities in ice. These approaches will help in understanding of scale dependency in sea ice deformation and in improving predictions of ice-covered seas in future and estimates related to climate warming.

## 285 Appendix A: Strain measures

Any strain tensor can be described in terms of deformation gradient  $\mathbf{F} = \mathbf{I} + \partial \mathbf{u}/\partial \mathbf{X}$ , where  $\mathbf{u} = \mathbf{u}(\mathbf{X},t)$  is a time-dependent displacement vector of a material point and the partial derivative taken with respect to its position vector  $\mathbf{X}$  in a reference configuration. By polar decomposition,  $\mathbf{F} = \mathbf{R}\mathbf{U}$ , where  $\mathbf{R}$  and  $\mathbf{U}$  are the rotation matrix and stretch tensor, respectively, describing the rigid body rotation and the actual deformation. Green-Lagrange strain tensor is then defined as (Gurtin, 1982)

$$\mathbf{E} = \frac{1}{2} \left( \mathbf{F}^T \cdot \mathbf{F} - \mathbf{I} \right) = \frac{1}{2} \left( \mathbf{U}^T \cdot \mathbf{R}^T \cdot \mathbf{R} \cdot \mathbf{U} - \mathbf{I} \right) = \frac{1}{2} \left( \mathbf{U}^T \cdot \mathbf{U} - \mathbf{I} \right),$$
 (A1)

since  $\mathbf{R}^T \cdot \mathbf{R} = \mathbf{I}$  for rotations. The last form of the this definition shows that  $\mathbf{E}$  is independent of rotations. The components of  $\mathbf{E}$  defined as a function of the displacement gradients are given by

$$E_{ij} = \frac{1}{2} \left( \frac{\partial u_i}{\partial x_j} + \frac{\partial u_j}{\partial x_i} + \frac{\partial u_k}{\partial x_i} \frac{\partial u_k}{\partial x_j} \right)$$
(A2)

and differ from the components of the infinitesimal strain tensor,  $\varepsilon$ , by the inclusion of quadratic terms, negligible for small deformations. Above we compare our results to previous large-scale estimates, that have typically used  $\varepsilon$ . For this it is practical to introduce unit extensions

$$l_{ii} = \sqrt{1 + 2E_{ii}} - 1$$
 and  $l_{xy} = \sin^{-1}\left(\frac{2E_{xy}}{\sqrt{(1 + 2E_{xx})(1 + 2E_{yy})}}\right)$  (A3)

which in the case of small deformations or deformations leading to normal strains only reduce to components  $\varepsilon_{ii}$  of  $\varepsilon$ . The principal values of 1 are given by

$$E_{1,2} = \frac{l_{xx} + l_{yy}}{2} \pm \sqrt{\left(\frac{l_{xx} - l_{yy}}{2}\right)^2 + l_{xy}^2}$$
 (A4)

and can be used to express the divergence, maximum shear, and the total deformation as

$$E_d = E_1 + E_2, E_s = \frac{1}{2}(E_1 - E_2) \text{and} E_t = \sqrt{E_d^2 + E_s^2}.$$
 (A5)

Similar operations can be applied in strain-rate analysis to derive  $\dot{E}_t$ , since the strain-rate tensor is derivable from the two strain tensors defined for two consecutive time steps within analyzed data.

## 305 Appendix B: Deformation estimate uncertainties



## B1 Error propagation and noise distributions

In addition to the error estimates presented in Uusinoka et al. (2025a), we estimate uncertainties in velocity and deformation by propagating position errors following Hutchings et al. (2012) and its ship-radar implementation in Oikkonen et al. (2017). In this framework, uncertainty depends on the per-fix position error  $\sigma_x$ , the time interval between sequential estimates T, the cell area A, and the ice velocity  $\mathbf{U}$ . Following Oikkonen et al. (2017), we'll consider the error estimates with U=0.01 m/s, which corresponds to 6 pixels with 10-min temporal intervals. Based on Uusinoka et al. (2025a, Supp. S2), the optical flow end-point error is approximately constant per frame and can be approximated as 0.06 pixels with this chosen value of U. With higher ice velocities, the relative error becomes smaller. Based on the synthetic data, a practical lower limit of applicability was suggested at 0.21 pixels displacement between frames, which corresponds to an ice velocity of U=0.0035 m/s with the temporal interval of 10 minutes. With a pixel size of 10 m and temporal interval of 10 min, we estimate our position error to  $\sigma_x=0.60$  m. For a cell with N vertices and area A, the area error is

$$\sigma_A = 2\sqrt{2} N \sqrt{A} \sigma_x, \tag{B1}$$

so that  $\sigma_A/A$  becomes negligible as  $A \gg 8N^2\sigma_x^2$ . For our quadrilateral cells, this condition becomes  $A \gg 46 \text{ m}^2$ . With our smallest nominal spatial scale of  $A = 400 \text{ m}^2$ , we have  $\frac{\sigma_A}{A} = 0.34$ . Furthermore, the strain rate error through propagation of position and time error is estimated as

$$\frac{\sigma_E}{E} = 2\left(4\frac{\sigma_x^2}{A} + 2\frac{\sigma_x^2}{\mathbf{U}^2 T^2} + \frac{\sigma_T^2}{T^2} + \frac{\sigma_A^2}{A^2}\right)^{1/2}.$$
(B2)

Similarly to previous implementations we assume  $\sigma_T/T \approx 0$ . With the highest available spatial (20 m) and temporal (10 min) resolutions, and for U=0.01 m/s, we have  $\frac{\sigma_E}{E}=0.75$ , which is comparable to the value of 0.58 presented in Oikkonen et al. (2017) for their estimates with 50 m and 10 min. Since  $\sigma_x^2/A$  and  $\sigma_A/A$  become negligible with larger spatial resolutions, the




$$\frac{\sigma_E}{E} \approx 2\sqrt{2} \frac{\sigma_x}{\mathbf{U}T}.\tag{B3}$$

With the larger nominal spatial scales we have  $\frac{\sigma_E}{E} \in [0.283, 0.094, 0.047, 0.016, 0.008, 0.004]$  for the temporal intervals of 10 min, 1 h, 3 h, 6 h, and 12 h, respectively. As shown in Uusinoka et al. (2025a), the estimate for relative errors for displacements decrease with displacement magnitude, similar values of  $\frac{\sigma_E}{E}$  can be achieved with lower ice velocities (e.g. for 0.0033 m/s  $\frac{\sigma_E}{E} \in [0.424, 0.141, 0.071, 0.024, 0.012, 0.006]$ ).

Since these estimates are based on the synthetic tests presented in Uusinoka et al. (2025a) and do not include all the possible noise artifacts cause by the field measurement conditions, we analyze the noise distributions of the MOSAiC data over an inactive region (shown in Figure B2a) in the radar coverage. We compare the deformation distributions within the same inactive region for two periods: during an active period (March case), and during a quiescent period with minimal motion across the radar coverage (February 17th to 22nd). Distributions are computed at the highest nominal scales of 10 min and 20 m.

Figure B1 shows that the deformation distribution is wider during the quiescent period. The mean deformation rate and thus the mean noise level is approximately  $10^{-3}$  h<sup>-1</sup> during both periods, which is approximately an order of magnitude smaller than the mean deformation estimates during an active period in Figure 5a. The slightly increased variability can be assumed due to the expectation that with large-scale displacement magnitudes the deep learning-based optical flow algorithm converges more confidently and thus produces tighter residuals off features. During the quiescent period, the gradient-based search of the optical flow algorithm explores more ambiguously and results in increased the standard deviation. The difference in the noise distributions combined with Eq. B1 suggests further that the scaling analysis can be assumed to be more reliable during active periods.

**Figure B1.** Noise distributions in an inactive region during active (March case) and quiescent (February) periods. The distributions are derived from the same region in the radar coverage. The distribution is slightly narrower during the active period and broader during quiescence with means deformation levels of  $10^{-3}$  h<sup>-1</sup> during both periods.

## B2 Scaling during a quiescent period


Since the noise pattern in the deformation estimates contain a checkerboard-like pattern, which could affect the observation of the lower bound with spatial scaling during winter events, we assess this possibility by repeating the mean-based scaling, first on a quiescent period between 17th to 22nd of February, and second on an inactive subregion within this period that exhibits the checkerboard artifact.

Figure B2a shows the cumulative deformation over the full quiescent period at highest available nominal resolutions. Since little deformation is observable during the quiescent period, the deformation field reveals a strong checkerboard pattern. The spatial scaling shown in Figure B2b exhibit the power-law behavior down to  $L=20~\mathrm{m}$  with  $R^2 \geq 0.95$ . Although no clear lower bound similarly to the active periods is observed, we note a slight increase in  $\langle \dot{E}_t(L,T) \rangle$  near  $L \sim 10^2~\mathrm{m}$  similarly to the spatial scaling of active periods, but this does not produce a clear scale break. The spatial scaling also has a similar slope

Figure B2. Cumulative total deformation rate over the quiescent period between 17th and 22nd of February at the nominal resolutions of 10 min and 20 m over a) the full radar coverage and c) on the inactive subregion. The inactive subregion is marked in a) as the rectangle. Spatial scaling of the quiescent period over b) the full radar coverage and d) on the inactive subregion. The power law fits all scales retained with  $R^2 \ge 0.95$  indicating that a similar lower-bound is not observed as during the active periods.

as the higher spatial scales during an active period. Since it is impossible to suggest this slight increase to result from the small

deformations or the checkerboard pattern, in Figure B2c-d we then analyze an inactive subregion of the radar coverage within

the same period with the highest available temporal resolutions to isolate noise structure. When we compute scaling over this

subregion, the fits for different T are nearly coinciding. The scaling can be described by a single power law across all L and

here no clear deviations can be seen from the power-law fit. Over this subregion, the slope of the fits is smaller than over the

whole area of radar coverage.

Although these error estimates and data checks suggest reliability in the main results of this paper, we note that during an

active period the checkerboard pattern noise is intensified around the deformation features as seen in Figure 4. To fully explore

the robustness of the lower bound around  $L \sim 10^2$  m, other datasets and deformation detection tools need to be utilized.

Code and data availability. The source code of the deformation detection method is archived in Zenodo (Uusinoka, 2024). The ship radar

raw data is available in PANGAEA (Krumpen et al., 2021). The time series and mean-based scaling data can be requested from Matias

Uusinoka.




Competing interests. At least one of the (co-)authors is a member of the editorial board of The Cryosphere. The authors have no other

competing interests to declare.

Author contributions. Conceptualization: MU, AP, JH

Formal analysis: MU

Data curation: MU, JH, ML

Investigation: MU, AP, JH, JÅ

Funding acquisition: JH, AP

Methodology: MU, AP, JH, JÅ, ML

**Project Administration:** AP

Resources: AP, JH


Software: MU

Writing - Original Draft: MU

Writing - Review & Editing: MU, AP, JH, JÅ, ML

Acknowledgements. MU and AP are grateful for financial support from the Research Council of Finland through the project (347802)

DEMFLO: Discrete Element Modeling of Continuous Ice Floes and Their Interaction. Contribution of JH was covered by the the European

Union's Horizon 2020 research and innovation programme under grant agreement No 101003826 via project CRiceS (Climate Relevant

interactions and feedbacks: the key role of sea ice and Snow in the polar and global climate system). JÅ was supported by the NOCOS DT project, funded by the Nordic Council of Ministers. All authors wish to acknowledge CSC – IT Center for Science, Finland, for computational resources under the project (2006428) DEMFLO, and the international Multidisciplinary drifting Observatory for the Study of the Arctic Climate (MOSAiC) project with the tag MOSAiC20192020 and the Project\_ID:AWI\_PS122\_00 for providing the ship radar data. Finally, we thank the cruise participants, ship's crew and logistics support as well as everyone else who contributed to the realization of MOSAiC (Nixdorf et al., 2021).

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
