# Peer review of "Scale invariance in kilometer-scale sea ice deformation"

_EGUsphere, 2025_

## Referee Comment (RC2)

**Comment on egusphere-2025-311**

*Damien Ringeisen*

In this manuscript, the author analyze sea ice radar images at high temporal and spatial resolution from the MOSAiC expedition. Using a novel method (described in a previous publication), they extract the deformation rates at unprecedented resolution and perform a scaling analysis of these deformation rates. The results show that the scaling laws of sea ice deformation break around $10^2$ m. Below this limit, deformation seems not scale and resemble the uniform viscous deformation of a summer sea ice field where the sea ice internal stress is negligible. This results seem to indicate that sea ice rheological model used at larger scales cannot represent the small scale dynamics below that limit.

I am really excited to the see that progress is being done on analysis of the dynamics of sea ice using data from sea ice radars. This is a very interesting study with a huge potential given the novelty of the dataset, I have however a few comments that I think should be addressed before publication.

**1. General comments**

I have tried to apply a simple MCC method on sea ice radar images during an expedition prior to MOSAiC, without luck. I am happy to see advancement in the treatment of these images, but I am wondering what is the estimation of the error on the deformation rates, and then what would be the error on the scaling exponents. I have searched the method paper to try to find some pointers but did not find a clear idea. In any case, a description of errors would strengthen this paper, maybe the authors can make an analysis inspired or similar as the one shown in Bouchat and Tremblay (2020).

I am questioning myself about the time selection used for the results. The fact that there are periods with no or very small deformation is very interesting, and confirms the concept of scaling: we know that sea ice kind of deforms constantly on the global scale, the MOSAiC expedition was then probably situated in between zones of high deformation, in the viscous/ductile zone. Then, active zones sometimes go through the sea ice radar range, and you then see a lot of deformation. What would be the results of the scaling analysis if the whole timeseries is used? If only the period with small deformation? I think this is necessary and interesting, and would strengthen the paper,

I have always been wondering if the scaling of shear and divergence would be different at very high resolution like here. My reasoning is that as you now see directly the rigid sea ice body floes, and not only the dynamics of the sea ice aggregate, there might be a difference as floes do not converge. Have you looked and seen a difference?

On significance of this study, I would also add that scaling is good metric for sea ice rheological models, but it is not the only one. A model can have a good scaling law but do not represent other sea ice properties well (see the SIREX 1 and 2 papers in refs). There is so much potential in the new dataset presented here, I feel the authors are barely scratching the surface here. I know that this is out of scope of this paper, I would love to see more.

**2. Specific comments**

**1. Introduction**

I was bit surprised by the introduction, the author include a lot of description of the novelty of their study and results already within the second paragraphs and onward (Our work, we describe, we employ...).

The paragraphs here seem to follow a structure: {1: New in our study, including results to address a knowledge gap} {2: what was done before}

while the inverse is usually used: {1: what was done before } {2: where is the knowledge gap}; and then in the last introduction paragraph {3: how we address the knowledge/method gap described above}

This is not necessarily wrong but it feels strange to read when we are used to a more traditional structure. I do not think that new/different ways of writing papers should necessarily be cast away, I wanted to just notify the authors, so they can decide if they want to keep it as it is.

**2. Methods**

*p4, L94:* "*To further ensure the robustness against the high-frequency noise in the data, described in Uusinoka et al. (2025), we chose 10-minute intervals between sequential images for sufficient displacements. We use 24-hour trajectories to avoid data loss due to artificial rotations in the radar images.*" I am really confused by this, do you use 24h trajectories or 10min trajectories? this is not clear to me.

**Results**

*p5, L121:* do you mean: "*Figure 2 shows deformation rate averaged for the 10 km×10 km sea ice area around ...* "

*figure 2 and 3:* I was wondering if the average deformation rate is the best to show here, wouldn't the maximum deformation be more adapted, we are interested in the localization of the deformation rate?

*p6, figure 3:* The units are missing on the left side of the plots, I would recommend to put the season names as subplot titles.

*p6, figure 3:* Is the "core" vertical scale (not the extensions for very high or small values) is at the same scale in all the panels? please check.

*p7, figure 4:* a video of the sea ice deformation (shear and divergence) as supplementary material would be very interesting to understand better the dynamics and what is happening here. I especially see that for November, there is an alternance of divergence and convergence withing the studied period. One could add the the radar field as well. Also showing how the deformation looks like at a single 10minute period would be nice as it is what is used for the scaling analysis.

*p7, figure 4:* Zooming on the figure, we can clearly see an grid like pattern in the deformation which I assume arises from the deformation calculations and uncertainty or noise. We see the same in our recent publication (Plante et al., 2025). I feel this should be described and its impact on results assessed.

*p8, L156:* "*Figure 5a depicts how the scaling exponent β behaved in the case of observations made over various spatial and temporal scales, L and T, respectively.*" respectively of what? something is missing here.

*p8, figure 5:* The figure caption should state that the deformation rate for B are for the 10 minutes interval

**Discussion**

*p11, L229:* like said in the general comment, I feel there is not only sea ice scaling that need to be addressed for sea ice models.

**Conclusions**

*p12, L246:* "*this paper*" I would suggest "*Our analysis of high resolution MOSAiC deformation data...*" or something like that to be more precise.

**3. References**

Bouchat, A., & Tremblay, B. (2020). Reassessing the quality of sea-ice deformation estimates derived from the RADARSAT Geophysical Processor System and its impact on the spatiotemporal scaling statistics. *Journal of Geophysical Research: Oceans*, 125, e2019JC015944. https://doi.org/10.1029/2019JC015944

Bouchat, A., Hutter, N., Chanut, J., Dupont, F., Dukhovskoy, D., Garric, G., et al. (2022). Sea Ice Rheology Experiment (SIREx): 1. Scaling and statistical properties of sea-ice deformation fields. *Journal of Geophysical Research: Oceans*, 127, e2021JC017667. https://doi.org/10.1029/2021JC017667

Hutter, N., Bouchat, A., Dupont, F., Dukhovskoy, D., Koldunov, N., Lee, Y. J., et al. (2022). Sea Ice Rheology Experiment (SIREx): 2. Evaluating linear kinematic features in highresolution sea ice simulations. *Journal of Geophysical Research: Oceans*, 127, e2021JC017666. https://doi.org/10.1029/2021JC017666

Plante, M., Lemieux, J.-F., Tremblay, L. B., Bouchat, A., Ringeisen, D., Blain, P., Howell, S., Brady, M., Komarov, A. S., Duval, B., Yakuden, L., and Labelle, F.: A sea ice deformation and rotation rate dataset (2017–2023) from the Environment and Climate Change Canada automated sea ice tracking system (ECCC-ASITS), Earth Syst. Sci. Data, 17, 423–434, https://doi.org/10.5194/essd-17-423-2025, 2025.

---

## Author Comment (AC1)

**Response to the Reviewer Comments on Manuscript**
**"Scale invariance in kilometer-scale sea ice deformation"**
**"Reviewer #1"**

*corresponding author: Matias Uusinoka (matias.uusinoka@aalto.fi)*

April 24, 2025

We sincerely thank the reviewer for their important and constructive comments regarding the potential impact of radar noise on our observation on sea ice deformation ceasing to exhibit scale invariance below $\sim 10^2$ m. We fully agree that any new observation challenging the status quo requires thorough investigation on any possible errors, and we appreciate the opportunity to highlight the checks we performed to rule out noise as the source of the observed lower-limit cutoff.

**General comments**

According to our understanding, ship radar data includes two different kind of noise which impact on the quality of the derived results. Firstly, in some situations, high frequency ($< 30$ sec) speckle can be detected from raw images. Those are probably due to waves in pack ice which cause vertical motion of ridges and edges of leads. Consequently, location and intensity of the strong backscattering objects are changing. Motion related to those events is not detectable with the radar system and in this analysis, we have been filtered that noise and consider only meter scale displacements. Moreover, wave induced speckle is assumed to be very small during the these selected periods.

The other sources of noise are due vibration of the radar antenna, shaking of the entire ship and interference with other radars. These effects were particular evident during the March deformation case and thus we paid attention to improve quality to data.

We note that our initial suspicion was indeed that the high-frequency noise in the MOSAiC radar data might introduce some spurious effects below some threshold scale. Consequently, a long time was devoted to attempting to invalidate this lower-limit scaling collapse. We, thus, examined our results from multiple angles. In (1), we measured the accuracy of our deep-learning-based optical flow by applying it to synthetic sea ice motion fields. Because these fields have known ground-truth displacements, we could quantify the end-point error over a range of motion types and magnitudes. The neural network had been trained originally on data sets with comparatively larger and more dynamic displacements, so our error estimates are assumed conservative when evaluating fine-scale motion in sea ice cover. In these artificial tests, no cutoff similar to the $\sim 10^2$ m could be observed. The algorithm works well when tracking small pixel displacements.

To provide further robustness against the unique high-frequency radar noise of the MOSAiC data, trained the neural network first with Gaussian noise over the training data and then specifically finetuned the model by applying noise maps sampled during stationary conditions in the MOSAiC data to apply them on top of the training set. This step is assumed to provide additional robustness against the radar noise in the MOSAiC data.

In the Supporting Information of (1), we evaluated the signal-to-noise ratio by following (2) and found that even at the finest scales used in our analysis, the signal remains sufficiently above the estimated noise. The strain fields over active leads yield a signal-to-noise ratio well above unity with no spatial or temporal

averaging applied. We note that in the scaling analysis in the current article, we only considered highly active periods where the signal-to-noise ratio was ensured to be constantly sufficiently high.

To consider other data sets with very different noise maps, we analyzed data from R/V Kronprins Haakon, which lacked the same high-frequency noise artifacts. Despite those data being limited to a few time windows lasting less than 12 hours, the scaling analysis for pack ice in summer surprisingly displayed behavior that seems to support the MOSAiC record (Fig 1).

[Figure]

Fig. 1. Comparison between the MOSAiC data and ship radar data from R/V Kronprins Haakon. Both data sets are from pack ice conditions, but the R/V Kronprins Haakon data was gathered during August 2023, which resulted in the higher deformation rates. The overlap between the results suggests some possible physical connection between the ice behavior. An example of the deformation map can be found in (1).

One piece of evidence is that the July case, which has radar noise levels comparable to the winter period, does not exhibit the scale collapse. If noise alone were driving the cutoff, we would expect to see a similar breakdown in scaling throughout the MOSAiC data set. Instead, we find the cutoff consistently in winter pack ice and consistently absent in July conditions, which suggests that the scaling collapse would be tied to some mechanical processes rather than measurement artifacts. In the winter cases, we furthermore observe that the typical width of fracture zones or leads is on the order of hundred to few hundred meters, which overlaps with the magnitude of the scaling collapse.

We also note that a single-day analysis, focusing on either entirely quiescent or entirely active phases, shows a scaling fit down to smaller scales, whereas combining multiple episodes of intermittent deformation causes the collapse near $10^2$ m also supports the view that the cutoff is not merely random noise. In that sense, the bulk statistics over an active two-week period detect transitions in activity that accumulate into a breakdown in the single power-law form below the size of the dominant deformation features.

We tested this with alternative approaches to scaling, such as coarse graining to confirm that the same cutoff behavior appears, albeit with differing values of the spatial scaling exponent.

We also note that we have ongoing work that expands this view of high-resolution ice dynamics observations to the multifractal aspects. In the multifractal analysis, we provide some support for these results, but also see that the behavior between the cases are highly different. Those results will, though, be published separately.

We agree with the reviewer that our observation on the scaling collapse requires further independent analysis. We hope our findings encourage additional campaigns, alternative methods, and fresh data to investigate whether a similar scale break appears in other regions or ice types. For the time being, our work with the synthetic-data tests, finetuning the neural network with the MOSAiC radar noise, signal-to-noise ratios, the disappearance of the scale collapse in the summer, the overlap with the R/V Kronprins Haakon ship radar data, and the physical consistency with the length scale of major fracture features gives us enough confidence in the result on the scaling collapse that we find it to be worth publishing. We will, however, revise the manuscript text to emphasize these points noise impacts. We hope that these clarifications address the reviewer's primary concern and suggests robustness of our conclusion.

**Specific comments**

1. *The paper in many places has sentences with grammar that I found hard to follow. A copy editor should be able to help here. I honestly stopped trying to correct them in my read. I will happily do this on a second draft of the paper.*

   We will go through the article sentence by sentence to try to resolve the grammatical erros and confusing sentences.

2. *The gray lines in figure 3 were hard to see in my print out.*

   We will increase the color intensity for the grey lines in Figure 3.

3. *The units of each term in figure A2 do not match, or is it that I am rusty in Einstein notation (I admit I prefer reading vector notation).*

   The terms in Eq. A2 for the Green-Lagrangian strain tensor should be correct. Following the Einstein notation, the part of $\sum_k$ is typically omitted from the full form of

$$E_{ij} = \frac{1}{2} \left( \frac{\partial u_i}{\partial x_j} + \frac{\partial u_j}{\partial x_i} + \sum_k \frac{\partial u_k}{\partial x_i} \frac{\partial u_k}{\partial x_j} \right). \tag{1}$$

**REFERENCES**

[1] M. Uusinoka, J. Haapala, and A. Polojärvi, "Deep learning-based optical flow in fine-scale deformation mapping of sea ice dynamics," *Geophysical Research Letters*, vol. 52, no. 2, p. e2024GL112000, 2025.

[2] A. Bouchat and B. Tremblay, "Reassessing the quality of sea-ice deformation estimates derived from the radarsat geophysical processor system and its impact on the spatiotemporal scaling statistics," *Journal of Geophysical Research: Oceans*, vol. 125, no. 8, p. e2019JC015944, 2020.

---

## Author Comment (AC2)

**Response to the Reviewer Comments on Manuscript**
**"Scale invariance in kilometer-scale sea ice deformation"**
**"Reviewer #2"**

*corresponding author: Matias Uusinoka (matias.uusinoka@aalto.fi)*

April 24, 2025

We would like to thank the reviewer for the encouraging comments and the positive assessment of our dataset's potential for exploring small-scale sea ice dynamics. We share the reviewer's enthusiasm for the potential in finding new insights from the fine-resolution deformation maps and look forward to expanding our analysis in future work. Below, we provide responses to the reviewer comments one by one.

**General comments**

We would like to thank the reviewer for the encouraging comments and the positive assessment of our dataset's potential for exploring small-scale sea ice dynamics. We share the reviewer's enthusiasm for the potential in finding new insights from the fine-resolution deformation maps and look forward to expanding our analysis in future work. Below, we provide responses to the reviewer comments one by one.

We initially shared the reviewer's concerns about uncertainty in the deformation estimates. To address this, we incorporated an error assessment approach similar to the one proposed in (1). In the methods paper (2) and its Supporting Information, we show the end-point errors (EPE) computed using synthetic ice motion fields, where we know ground-truth displacements at each pixel. We also quantify signal-to-noise ratios for the real radar-derived strain rates. Even at the highest spatiotemporal resolutions, these ratios remain comfortably above unity in areas undergoing deformation, implying that the extracted signals are not buried by radar noise. We will add further clarifications in the revised manuscript on how these uncertainties translate into error in the strain estimates.

Regarding the choice to focus on time periods of relatively large deformation, we are very happy about the idea of the reviewer to show a continuous time series over the whole MOSAiC data set and agree that it is interesting to see how the scaling analysis holds for the entirety of the MOSAiC measurements. We have performed exactly this type of a broad scaling analysis, where scaling was performed with a sliding window over the whole winter. The results of this analysis, though, are planned to be published in another paper later. Additionally, since this paper is first to publish results over these scales with new results, we wanted to concentrate on highly active deformation periods where the signal-to-noise ratios can be ensured to be consistently high enough.

The reviewer suggests comparing the scaling of divergence and shear separately at these small scales, given the possibility that individual floes might not converge as much as they shear. We have performed only some small analysis at this question but hope to expand on this and integrate such this with a discrete element modeling framework in future work. Additionally, we'd like to use multifractal analysis to for this comparison to better describe their individual characteristics.

As the reviewer rightly points out, scaling alone cannot capture all the complexities of ice mechanical behavior. Our results expand one of these properties to smaller scales to provide empirical observations on the varying behavior of these methods at unexplored scales. We will add a comment in the article that while

the fractal properties are a major properties of large-scale ice covers, there are many properties that current models cannot capture yet.

**Specific comments**

1.1 *Comments on the introduction*

We thank the reviewer for their kind words about the introduction. The current text uses a structure in which we highlight our study's novelty early on, without forgetting the state-of-the-art review. Although this way departs from the traditional style, we felt it would help the reader in understanding what our paper is about and to high-resolution radar dataset and the results from the outset.

2.1 *p4, L94: am really confused by this, do you use 24h trajectories or 10min trajectories? this is not clear to me.*

We use 24-hour trajectories that are generated based on sequential radar images with 10-minute intervals. We will clarify this further in the revised article.

3.1 *p5, L121: do you mean: "Figure 2 shows deformation rate averaged for the 10 km × 10 km sea ice area around . . . "*

This is what was tried to communicate. We will change the "*records for the 10 km × 10 km*" to "*averaged over the 10 km × 10 km*".

3.2 *figure 2 and 3: I was wondering if the average deformation rate is the best to show here, wouldn't the maximum deformation be more adapted, we are interested in the localization of the deformation rate?*

We use the average deformation rate for the time series as this supports the analysis of the first moment fractal dimensions. Using the maximum deformation rates will suit our future analysis concentrating on multifractals, where the higher moments will provide better description on the scaling of the extreme values.

3.3 *p6, figure 3: The units are missing on the left side of the plots, I would recommend to put the season names as subplot titles.*

Although originally trying a slightly unconventional visualization of the data, we agree with the reviewer that having the case names in subplot titles will make the figure easier to read. We will change this.

3.4 *p6, figure 3: Is the "core" vertical scale (not the extensions for very high or small values) is at the same scale in all the panels? please check.*

Different vertical limits were originally used to highlight the individual deformation behavior during the different cases. We find the reviewers approach to make the figure more easily interpretable and will modify the figure to have identical core vertical scales for all winter cases. For July cases we double the scale on vertical axis due to the intensity of deformation but this will be pointed out in the figure caption.

3.5 *p7, figure 4: a video of the sea ice deformation (shear and divergence) as supplementary material would be very interesting to understand better the dynamics and what is happening here. I especially see that*

*for November, there is an alternance of divergence and convergence withing the studied period. One could add the the radar field as well. Also showing how the deformation looks like at a single 10minute period would be nice as it is what is used for the scaling analysis.*

We are preparing a follow-up article where the November and other sea ice deformation events are analyzed in details. The animation of sea ice deformation will be included as to that manuscript. Examples of a 10-minute deformation map are shown in (2), which we will point out in the revised version of our paper.

3.6 *p7, figure 4: Zooming on the figure, we can clearly see an grid like pattern in the deformation which I assume arises from the deformation calculations and uncertainty or noise. We see the same in our recent publication (Plante et al., 2025). I feel this should be described and its impact on results assessed.*

We will elaborate on this in the revised paper. Concerning the grid pattern that occurs in regions of very small deformation and where radar noise partially obscures the signal. According to (3), these features may appear when the nominal resolution is not sufficient to capture the gradients in sea ice displacement.

Concerning the scaling analysis, the deformation rate distributions over areas that only contain this grid-like patterns contain only values smaller than the mean of the whole deformation field and are roughly two to three magnitudes lower than the identified large deformation features. The increased values over quiescent areas might decrease the spatial scaling exponent, $\beta$, as the field becomes slightly more homogeneous but should not affect the result of scaling collapse.

We find it more likely that there might be smaller deformation features that cannot be distinguished from the radar images, which would result in the loss of scale invariance. As this cannot be determined from the radar data, we suggest in the revised article that the lower bound for scale invariance should be sought also in other data sets and methods in future work.

3.7 *p8, L156: "Figure 5a depicts how the scaling exponent $\beta$ behaved in the case of observations made over various spatial and temporal scales, L and T, respectively." respectively of what? something is missing here.*

The word respectively here links spatial and temporal scales to L and T. We will reword the sentence.

3.8 *p8, figure 5: The figure caption should state that the deformation rate for B are for the 10 minutes interval*

This is a good catch by the reviewer. We will add this mention into the caption of Figure 5.

4.1 *p11, L229: like said in the general comment, I feel there is not only sea ice scaling that need to be addressed for sea ice models.*

We will extend the discussion in the revised paper to point out that while scaling and fractal analysis are a major component of model validation in recent years, it is most certainly not the only one.

5.1 *p12, L246: "this paper" I would suggest "Our analysis of high resolution MOSAiC deformation data..." or something like that to be more precise.*

We will change the wording from "this paper" to the revised version of the article.

REFERENCES

[1] A. Bouchat and B. Tremblay, "Reassessing the quality of sea-ice deformation estimates derived from the radarsat geophysical processor system and its impact on the spatiotemporal scaling statistics," *Journal of Geophysical Research: Oceans*, vol. 125, no. 8, p. e2019JC015944, 2020.

[2] M. Uusinoka, J. Haapala, and A. Polojärvi, "Deep learning-based optical flow in fine-scale deformation mapping of sea ice dynamics," *Geophysical Research Letters*, vol. 52, no. 2, p. e2024GL112000, 2025.

[3] M. Plante, J.-F. Lemieux, L. B. Tremblay, A. Bouchat, D. Ringeisen, P. Blain, S. Howell, M. Brady, A. S. Komarov, B. Duval, *et al.*, "A sea ice deformation and rotation rates dataset (2017–2023) from the environment and climate change canada automated sea ice tracking system (eccc-asits)," *Earth System Science Data Discussions*, vol. 2024, pp. 1–19, 2024.

---

## Author Response (AR1)

**Response to the Reviewer Comments on Manuscript "Scale invariance in kilometer-scale sea ice deformation"**

corresponding author: Matias Uusinoka (matias.uusinoka@aalto.fi)

May 15, 2025

We express our sincere gratitude for the insightful and constructive comments and suggestions provided by the reviewers. We have tried to address all the comments. In the revised manuscript, modifications and amendments are highlighted. In the following, we respond to the reviewer comments in a point-by-point manner.

**Reviewer #1**

**General comments**

1.1 My one concern is that we have not examined the accuracy of the Uunsinoka et al. (2025) tracking method at all scales. Uusinoka (2025) figure 1b3 shows how noise in the RADAR imagery is manifest. This noise, which we assume is from waves and wind "wobble", appears to have a course grain structure that is of order 100m wide. I am guessing the dimensions of the course graining comparing to the scale in Figure 1. How does this impact the RAFT solution at 102 m (100m) length scales? Is the "surprising" result that scaling behavior disappears below 100m simply that you are observing noise between vectors at this scale. If you are sampling noise you would expect beta and alpha to go to zero. If this noise has a particular course grain structure that is imparted by the methodology, you would expect Lc to be constant across the analysis. Should noise be an issue, the main finding of the paper needs to be refocussed - because you are not finding a lower limit on scaling that is applicable to the modeling community.

According to our understanding, ship radar data includes two different kind of noise which impact on the quality of the derived results. Firstly, in some situations, high frequency (< 30 sec) speckle can be detected from raw images. Those are probably due to waves in pack ice which cause vertical motion of ridges and edges of leads. Consequently, location and intensity of the strong backscattering objects are changing. Motion related to those events is not detectable with the radar system and in this analysis, we have been filtered that noise and consider only meter scale displacements. Moreover, wave induced speckle is assumed to be very small during the these selected periods.

The other sources of noise are due vibration of the radar antenna, shaking of the entire ship and interference with other radars. These effects were particular evident during the March deformation case and thus we paid attention to improve quality to data.

We note that our initial suspicion was indeed that the high-frequency noise in the MOSAiC radar data might introduce some spurious effects below some threshold scale. Consequently, a long time was devoted to attempting to invalidate this lower-limit scaling collapse. We, thus, examined our results from multiple angles. In [1], we measured the accuracy of our deep-learning-based optical flow by applying it to synthetic sea ice motion fields. Because these fields have known ground-truth displacements, we could quantify the end-point error over a range of motion types and magnitudes. The neural network had been trained originally on data sets with comparatively larger and more

dynamic displacements, so our error estimates are assumed conservative when evaluating fine-scale motion in sea ice cover. In these artificial tests, no cutoff similar to the  $\sim 10^2$  m could be observed. The algorithm works well when tracking small pixel displacements.

To provide further robustness against the unique high-frequency radar noise of the MOSAiC data, trained the neural network first with Gaussian noise over the training data and then specifically finetuned the model by applying noise maps sampled during stationary conditions in the MOSAiC data to apply them on top of the training set. This step is assumed to provide additional robustness against the radar noise in the MOSAiC data.

In the Supporting Information of [1], we evaluated the signal-to-noise ratio by following [2] and found that even at the finest scales used in our analysis, the signal remains sufficiently above the estimated noise. The strain fields over active leads yield a signal-to-noise ratio well above unity with no spatial or temporal averaging applied. We note that in the scaling analysis in the current article, we only considered highly active periods where the signal-to-noise ratio was ensured to be constantly sufficiently high.

To consider other data sets with very different noise maps, in the The Cryosphere discussion forum we should analyzed data from R/V Kronprins Haakon, which lacked the same high-frequency noise artifacts. Despite the data being limited to a few time windows lasting less than 12 hours, the scaling analysis for pack ice in summer surprisingly displayed behavior that seems to support the MOSAiC record.

One piece of evidence is that the July case, which has radar noise levels comparable to the winter period, does not exhibit the scale collapse. If noise alone were driving the cutoff, we would expect to see a similar breakdown in scaling throughout the MOSAiC data set. Instead, we find the cutoff consistently in winter pack ice and consistently absent in July conditions, which suggests that the scaling collapse would be tied to some mechanical processes rather than measurement artifacts. In the winter cases, we furthermore observe that the typical width of fracture zones or leads is on the order of hundred to few hundred meters, which overlaps with the magnitude of the scaling collapse. We also note that a single-day analysis, focusing on either entirely quiescent or entirely active phases, shows a scaling fit down to smaller scales, whereas combining multiple episodes of intermittent deformation causes the collapse near  $10^2$  m also supports the view that the cutoff is not merely random noise. In that sense, the bulk statistics over an active two-week period detect transitions in activity that accumulate into a breakdown in the single power-law form below the size of the dominant deformation features.

We tested this with alternative approaches to scaling, such as coarse graining to confirm that the same cutoff behavior appears, albeit with differing values of the spatial scaling exponent.

We also note that we have on-going work that expands this view of high-resolution ice dynamics observations to the multifractal aspects. In the multifractal analysis, we provide some support for these results, but also see that the behavior between the cases are highly different. Those results will, though, be published separately.

We agree with the reviewer that our observation on the scaling collapse requires further independent analysis. We hope our findings encourage additional campaigns, alternative methods, and fresh data to investigate whether a similar scale break appears in other regions or ice types. For the time being, our work with the synthetic-data tests, finetuning the neural network with the MOSAiC radar noise, signal-to-noise ratios, the disappearance of the scale collapse in the summer, the overlap with the R/V Kronprins Haakon ship radar data, and the physical consistency with the length scale of major fracture features gives us enough confidence in the result on the scaling collapse that we find it to

be worth publishing. We have, however, revised the manuscript text to emphasize these points noise impacts in the discussion (L215). We hope that these clarifications address the reviewer's primary concern and suggests robustness of our conclusion.

1.2 The results could be combined with buoy analysis from the MOSAiC DN to extend to 1000km scales (which is outside of the scope of this paper, but I would like to see done).

We appreciate the reviewer's suggestion to combine our radar-based deformation fields with the buoy data. We agree that integrating the two data sets would yield a more complete picture of sea-ice dynamics and would be very happy to pursue this type of a synthesis.

**Specific comments: Text**

2.1 The paper in many places has sentences with grammar that I found hard to follow. A copy editor should be able to help here. I honestly stopped trying to correct them in my read. I will happily do this on a second draft of the paper.

We have gone through the article trying to resolve some of the grammatical errors and confusing sentences. We are also happy to correct any confusing sentences left in the article as noted by the reviewer.

2.2 The gray lines in figure 3 were hard to see in my print out.

We have increased the color intensity for the grey lines in Figure 3.

2.3 The units of each term in figure A2 do not match, or is it that I am rusty in Einstein notation (I admit I prefer reading vector notation)

The terms in Eq. A2 for the Green-Lagrangian strain tensor should be correct. Following the Einstein notation, the part of  $\sum_k$  is typically omitted from the full form of

$$E_{ij} = \frac{1}{2} \left( \frac{\partial u_i}{\partial x_j} + \frac{\partial u_j}{\partial x_i} + \sum_k \frac{\partial u_k}{\partial x_i} \frac{\partial u_k}{\partial x_j} \right). \tag{1}$$

**Reviewer #2**

**General comments**

1.1 I have tried to apply a simple MCC method on sea ice radar images during an expedition prior to MOSAiC, without luck. I am happy to see advancement in the treatment of these images, but I am wondering what is the estimation of the error on the deformation rates, and then what would be the error on the scaling exponents. I have searched the method paper to try to find some pointers but did not find a clear idea. In any case, a description of errors would strengthen this paper, maybe the authors can make an analysis inspired or similar as the one shown in Bouchat and Tremblay (2020).

We initially shared the reviewer's concerns about uncertainty in the deformation estimates. To address this, we incorporated an error assessment approach similar to the one proposed in [2]. In the methods paper [1] and its Supporting Information, we show the end-point errors (EPE) computed using synthetic ice motion fields, where we know ground-truth displacements at each pixel. We also quantify signal-to-noise ratios for the real radar-derived strain rates. Even at the highest spatiotemporal resolutions, these ratios remain consistently high in areas undergoing deformation, implying that the extracted signals are not buried by radar noise. We have added further clarifications in the revised manuscript on how these uncertainties translate into error in the strain estimates (L215).

1.2 I am questioning myself about the time selection used for the results. The fact that there are periods with no or very small deformation is very interesting, and confirms the concept of scaling: we know that sea ice kind of deforms constantly on the global scale, the MOSAiC expedition was then probably situated in between zones of high deformation, in the viscous/ductile zone. Then, active zones sometimes go through the sea ice radar range, and you then see a lot of deformation. What would be the results of the scaling analysis if the whole timeseries is used? If only the period with small deformation? I think this is necessary and interesting, and would strengthen the paper.

Regarding the choice to focus on time periods of relatively large deformation, we are very happy about the reviewer's idea to show a continuous time series over the whole MOSAiC data set and agree that it is interesting to see how the scaling analysis holds for the entirety of the MOSAiC measurements. We have performed exactly this type of a broad scaling analysis, where scaling was performed with a sliding window over the whole winter. The results of this analysis, though, are planned to be published in another paper later. Additionally, since this paper is first to publish results over these scales with new results, we wanted to concentrate on highly active deformation periods where the signal-to-noise ratios can be ensured to be consistently high enough, which we now mention in L160 in the revised manuscript.

1.3 I have always been wondering if the scaling of shear and divergence would be different at very high resolution like here. My reasoning is that as you now see directly the rigid sea ice body floes, and not only the dynamics of the sea ice aggregate, there might be a difference as floes do not converge. Have you looked and seen a difference?

The reviewer suggests comparing the scaling of divergence and shear separately at these small scales, given the possibility that individual floes might not converge as much as they shear. We have performed only some small analysis of this sort but hope to expand on this and integrate this with a discrete element modeling framework in future work. Additionally, we'd like to use multifractal analysis to for this comparison to better describe their individual characteristics.

1.4 On significance of this study, I would also add that scaling is good metric for sea ice rheological models, but it is not the only one. A model can have a good scaling law but do not represent other sea ice properties well (see the SIREX 1 and 2 papers in refs). There is so much potential in the new dataset presented here, I feel the authors are barely scratching the surface here. I know that this is out of scope of this paper, I would love to see more.

As the reviewer rightly points out, scaling alone cannot capture all the complexities of ice mechanical behavior. Our results expand one of these properties to smaller scales to provide empirical observations on the varying behavior of these methods at unexplored scales. We have added a comment in the article (L254) that while the fractal properties are a major properties of large-scale ice covers, there are many properties that current models cannot capture yet.

**Specific comments: Text**

2.1 I was bit surprised by the introduction, the author include a lot of description of the novelty of their study and results already within the second paragraphs and onward (Our work, we describe, we employ...). The paragraphs here seem to follow a structure: 1: New in our study, including results to address a knowledge gap 2: what was done before while the inverse is usually used: 1: what was done before 2: where is the knowledge gap; and then in the last introduction paragraph 3: how we address the knowledge/method gap described above

This is not necessarily wrong but it feels strange to read when we are used to a more traditional structure. I do not think that new/different ways of writing papers should necessarily be cast away, I wanted to just notify the authors, so they can decide if they want to keep it as it is.

We thank the reviewer for their kind words about the introduction. The current text uses a structure in which we highlight our study's novelty early on, without forgetting the state-of-the-art review. Although this way departs from the traditional style, we felt it would help the reader in understanding what our paper is about and to high-resolution radar dataset and the results from the outset.

2.2 p4, L94: "To further ensure the robustness against the high-frequency noise in the data, described in Uusinoka et al. (2025), we chose 10-minute intervals between sequential images for sufficient displacements. We use 24-hour trajectories to avoid data loss due to artificial rotations in the radar images." I am really confused by this, do you use 24h trajectories or 10min trajectories? this is not clear to me.

We use 24-hour trajectories that are generated based on sequential radar images with 10-minute intervals. We have clarified this further in the revised article (L95).

2.3 p5, L121: do you mean: "Figure 2 shows deformation rate averaged for the 10 km×10 km sea ice area around ..."

This is what was tried to communicate. We've changeed the "records for the  $10 \text{ km} \times 10 \text{ km}$ " to "averaged over the  $10 \text{ km} \times 10 \text{ km}$ ".

2.4 figure 2 and 3: I was wondering if the average deformation rate is the best to show here, wouldn't the maximum deformation be more adapted, we are interested in the localization of the deformation rate?

We use the average deformation rate for the time series as this supports the analysis of the first moment fractal dimensions. Using the maximum deformation rates will suit our future analysis concentrating on multifractals, where the higher moments will provide better description on the scaling of the extreme values.

2.5 p6, figure 3: The units are missing on the left side of the plots, I would recommend to put the season names as subplot titles.

Although originally trying a slightly unconventional visualization of the data, we agree with the reviewer that having the case names in subplot titles will make the figure easier to read. We have now changed this.

2.6 p6, figure 3: Is the "core" vertical scale (not the extensions for very high or small values) is at the same scale in all the panels? please check.

Different vertical limits were originally used to highlight the individual deformation behavior during the different cases. We find the reviewers approach to make the figure more easily interpretable and have modified the figure to have identical core vertical scales for all winter cases. For July cases we doubled the scale on vertical axis due to the intensity of deformation but this has now been pointed out in the figure caption.

2.7 p7, figure 4: a video of the sea ice deformation (shear and divergence) as supplementary material would be very interesting to understand better the dynamics and what is happening here. I especially see that for November, there is an alternance of divergence and convergence withing the studied period. One could add the the radar field as well. Also showing how the deformation looks like at a single 10minute period would be nice as it is what is used for the scaling analysis.

We are preparing a follow-up article where the November and other sea ice deformation events are analyzed in details. The animation of sea ice deformation will be included as to that manuscript. Examples of a 10-minute deformation map are shown in [1], which we have pointed out in the revised version of our paper (L150).

2.8 p7, figure 4: Zooming on the figure, we can clearly see an grid like pattern in the deformation which I assume arises from the deformation calculations and uncertainty or noise. We see the same in our recent publication (Plante et al., 2025). I feel this should be described and its impact on results assessed.

We have elaborated on this in the revised paper. Concerning the grid pattern that occurs in regions of very small deformation and where radar noise partially obscures the signal. According to [3], these features may appear when the nominal resolution is not sufficient to capture the gradients in sea ice displacement.

Concerning the scaling analysis, the deformation rate distributions over areas that only contain this grid-like patterns contain only values smaller than the mean of the whole deformation field and are roughly two to three magnitudes lower than the identified large deformation features. The increased values over quiescent areas might decrease the spatial scaling exponent,  $\beta$ , as the field becomes slightly more homogeneous but should not affect the result of scaling collapse.

We find it more likely that there might be smaller deformation features that cannot be distinguished from the radar images, which would result in the loss of scale invariance. As this cannot be

determined from the radar data, we suggest in the revised article that the lower bound for scale invariance should be sought also in other data sets and methods in future work (L254).

2.9 p8, L156: "Figure 5a depicts how the scaling exponent  $\beta$  behaved in the case of observations made over various spatial and temporal scales, L and T, respectively." respectively of what? something is missing here.

The word respectively here links spatial and temporal scales to L and T. We have reworded the sentence.

2.10 p8, figure 5: The figure caption should state that the deformation rate for B are for the 10 minutes interval.

This is a good catch by the reviewer. We've added this mention into the caption of Figure 5.

2.11 p11, L229: like said in the general comment, I feel there is not only sea ice scaling that need to be addressed for sea ice models.

We've extended the discussion in the revised paper (L254) to point out that while scaling and fractal analysis are a major component of model validation in recent years, it is most certainly not the only one.

2.12 p12, L246: "this paper" I would suggest "Our analysis of high resolution MOSAiC deformation data..." or something like that to be more precise.

We've changeed the wording from "this paper" to the revised version of the article.

**REFERENCES**

- [1] M. Uusinoka, J. Haapala, and A. Polojärvi, "Deep learning-based optical flow in fine-scale deformation mapping of sea ice dynamics," *Geophysical Research Letters*, vol. 52, no. 2, p. e2024GL112000, 2025.
- [2] A. Bouchat and B. Tremblay, "Reassessing the quality of sea-ice deformation estimates derived from the radarsat geophysical processor system and its impact on the spatiotemporal scaling statistics," *Journal of Geophysical Research: Oceans*, vol. 125, no. 8, p. e2019JC015944, 2020.
- [3] M. Plante, J.-F. Lemieux, L. B. Tremblay, A. Bouchat, D. Ringeisen, P. Blain, S. Howell, M. Brady, A. S. Komarov, B. Duval, *et al.*, "A sea ice deformation and rotation rates dataset (2017–2023) from the environment and climate change canada automated sea ice tracking system (eccc-asits)," *Earth System Science Data Discussions*, vol. 2024, pp. 1–19, 2024.

---

## Referee Report (RR1)

**Comment on egusphere-2025-311 - Round 2**

**Damien Ringeisen**

I thank the authors for their answers to my comments and questions. I am quite satisfied with the answers and do not have additional major comments.

For two of my comments, the authors say they will publish additional papers with the results, I understand the authors choice to keep this paper focused on a single result.

**Specific comments**

- 1.1 I have tried to apply a simple MCC method on sea ice radar images during an expedition prior to MOSAiC, without luck. I am happy to see advancement in the treatment of these images, but I am wondering what is the estimation of the error on the deformation rates, and then what would be the error on the scaling exponents. I have searched the method paper to try to find some pointers but did not find a clear idea. In any case, a description of errors would strengthen this paper, maybe the authors can make an analysis inspired or similar as the one shown in Bouchat and Tremblay (2020).
  - We initially shared the reviewer's concerns about uncertainty in the deformation estimates. To address this, we incorporated an error assessment approach similar to the one proposed in [2]. In the methods paper [1] and its Supporting Information, we show the end-point errors (EPE) computed using synthetic ice motion fields, where we know ground-truth displacements at each pixel. We also quantify signal-to-noise ratios for the real radar-derived strain rates. Even at the highest spatiotemporal resolutions, these ratios remain consistently high in areas undergoing deformation, implying that the extracted signals are not buried by radar noise. We have added further clarifications in the revised manuscript on how these uncertainties translate into error in the strain estimates (L215).
    - Thanks for this addition, it addresses by comment, I would only recommend writing "
      [...] and presented in Uusinoka et al. (2025, Supp. S2) –demonstrate that localized [...]"
      to help the reader find the information, using \citet[][Supp. S2]{bibkey}
- Following comment 2.3 of reviewer 1, I also wondered about the units consistency of Equation A2 with the Einstein notation. After careful reading, I think the notation and units are correct because  $\vec{u}$  is not the *velocity* (as one would expect from the u usual convention) but the *displacement* vector in [m], hence  $\frac{\partial u_i}{\partial x_j}$  is unitless, and so is  $\frac{\partial u_k}{\partial x_i} \frac{\partial u_k}{\partial x_j}$ . I am not sure your answer points this out. However I am not sure how one would do to compute directly the strain rate by using the velocity instead of displacement.

---

## Author Response (AR2)

**Response to the Reviewer Comments on Manuscript "Scale invariance in kilometer-scale sea ice deformation"**

corresponding author: Matias Uusinoka (matias.uusinoka@aalto.fi)

September 18, 2025

First, we'd like to apologize for the delay in submitting the revised manuscript and response letter. The delay was due to the authors' summer holidays occurring at different times, followed by the corresponding author completing and submitting their doctoral dissertation. In addition, the generation of new deformation data for the error estimates required considerable time.

We'd also like to thank the reviewers for their important constructive comments. All authors agree that the presented comments have strengthened the manuscript quite significantly. This is especially important given that the paper presents new observations where the reliability of the results is absolutely necessary. In the following, we've tried to address all the presented comments. In the revised manuscript, modifications are highlighted in red, with the exception of Appendix B, which is presented in standard font for readability.

Changes made to the revised form of the article:

- In Appendix B1 we provide the error propagation over different scales similarly to [1] and [2] based on the displacement estimate errors from [3]. We also provide estimation on the noise floor based on deformation distributions over a inactive subregion in the radar coverage during both an active and a quiescent period.
- In Appendix B2 we provide spatial scaling estimates during a quiescent period in both the full radar coverage as well as in a inactive subregion showing the checkerboard structure.
- We've applied the suggested text edits, softened the claim of the lower bound slightly for better robustness, and added a short mention of the complementary results from [4].

We hope that these edits sufficiently address the concerns of data quality. We find that the new appendix section strengthens the paper considerably.

**Reviewer #1**

**General comments**

We thank the reviewer for the constructive follow-up. We fully agree that a major issue with noisy field data always is distinguishing signal from noise. We've now expanded the manuscript with a new section in the appendix to address this. Since the observations and the methodology are new, we've softened the claims of the article slightly to highlight the fact that further observations are needed to verify the existence of such lower bound. To address the concerns about data quality at higher resolutions in the new appendix section, we

1) quantify propagated errors and the noise floor. We use the error-propagation framework of [1] and used by [2] with error estimates from our synthetic tests in [3]. This provides explicit estimates of relative strain-rate errors as a function of spatial and temporal scale similar to previous work.

- 2) compare noise distributions during quiescent and active periods. We estimate the deformation noise level to be in the order of ~ 10-3 h-1, which is roughly an order of magnitude smaller than mean deformation rates during active events. On top of the signal-to-noise perspective, we justify concentrating the analysis to active periods by showing that the assumed noise distribution is narrower and thus more predictable during active periods.
- 3) test scaling during a quiescent period both with full radar coverage as well as with an inactive subregion in the radar data that is dominated by the checkerboard. The scaling seems to remains power-law-like with  $R^2 \ge 0.95$  with a slight increase for  $L 10^2$  m. In the inactive subregion we do not see this increase.

We hope that these additions address the concern that noise could affect the scaling analysis. While we believe we cannot fully resolve the question without independent work with other datasets and deformation estimation methods, these error estimates are assumed to give a clearer picture of the presence of noise in the analysis. We've revised the main text slightly to highlight that future work using complementary data sources will be needed to fully establish the robustness of the lower bound at  $\sim 10^2$  m.

**Specific comments: Text**

2.1 Page 2, line 50: There is some logic missing in this sentence. The fact that the study area has major and smaller deformation periods does not relate to the first clause in the sentence "We present three key findings that are all important to account for in modeling of sea ice". This is a description of your data, not a general finding.

We've now rephrased this paragraph to better distinguish between data description and general findings.

2.2 Page 3, line 60: Hutchings et al. (2024) found a transition between 5 and 10km, which is close to the edge of the range you can resolve in your study. They also had a lower resolution limit of more than 100m. So your findings are not incompatible with those of Hutchings et al. (2024), you would need a larger region to course grain over to capture the 1-100km range where a transition at around 10km is apparent. I understand that you explain this better in the discussion of the paper (page 12, lines 236-245), but feel that on page 3 the reader could be mislead by your phrase.

The sentence has now been changed to emphasize the complementary nature of our paper and Hutchings et al. (2024) already in the introduction.

2.3 Line 71: "the radar signal"  $\rightarrow$  "a radar signal"

This has now been fixed.

2.4 Line 75: check punctuation, a full stop is doubled.

We've removed unnecessary punctuation to increase readability.

2.5 Page 8 line 168: Should  $T \ge 10 \text{ min } be T = 10 \text{ min}$ ?

We've now changed it to "T = 10 min". Thank you for noticing!

2.6 Figure 5: Explain what the dashed line is  $(L_c?)$  in the caption.

We've now added a description for the dashed line in Figure 5.

**Reviewer #2**

**Specific comments: Text**

2.1 Thanks for this addition, it addresses by comment, I would only recommend writing "[...] and presented in Uusinoka et al. (2025, Supp. S2) –demonstrate that localized [...]" to help the reader find the information, using \citet[][Supp. S2]{bibkey}}.

This is a good point and something we should've already added earlier. Thank you for pointing this out!

2.2 Following comment 2.3 of reviewer 1, I also wondered about the units consistency of Equation A2 with the Einstein notation. After careful reading, I think the notation and units are correct because  $\mathbf{u}$  is not the velocity (as one would expect from the  $\mathbf{u}$  usual convention) but the displacement vector in [m], hence  $\frac{\partial u_i}{\partial x_j}$  is unitless, and so is  $\frac{\partial u_k}{\partial x_i} \frac{\partial u_k}{\partial x_j}$ . I am not sure your answer points this out. However I am not sure how one would do to compute directly the strain rate by using the velocity instead of displacement.

To encompass the reader better, we've added a small extension to the verbal description of strain tensor to highlight that  $\mathbf u$  is the displacement vector: "The components of  $\mathbf E$  defined as a function of the displacement gradients are given by ...".

**REFERENCES**

- [1] J. Hutchings, P. Heil, A. Steer, and W. Hibler III, "Subsynoptic scale spatial variability of sea ice deformation in the western weddell sea during early summer," *Journal of Geophysical Research: Oceans*, vol. 117, no. C1, 2012.
- [2] A. Oikkonen, J. Haapala, M. Lensu, J. Karvonen, and P. Itkin, "Small-scale sea ice deformation during n-ice 2015: From compact pack ice to marginal ice zone," *Journal of Geophysical Research: Oceans*, vol. 122, no. 6, pp. 5105–5120, 2017.
- [3] M. Uusinoka, J. Haapala, and A. Polojärvi, "Deep learning-based optical flow in fine-scale deformation mapping of sea ice dynamics," *Geophysical Research Letters*, vol. 52, no. 2, p. e2024GL112000, 2025.
- [4] M. Uusinoka, A. Savard, J. Åström, J. Haapala, and A. Polojärvi, "Threshold domain sizes for multifractality in sea ice deformation," *Geophysical Research Letters*, vol. 52, no. 16, p. e2025GL116833, 2025.